# Effectiveness of the Med Safety mobile application in improving adverse drug reaction reporting by healthcare professionals in Uganda: a protocol for a pragmatic cluster-randomised controlled trial

Ronald Kiguba ![ORCID],[1] Norah Mwebaza,[1] Ronald Ssenyonga ![ORCID],[2]
Helen Byomire Ndagije,[3] Victoria Nambasa,[3] Cordelia Katureebe,[4]
Kenneth Katumba,[5] Phil Tregunno,[6] Kendal Harrison,[6] Charles Karamagi,[7]
Kathryn A Scott,[8] Munir Pirmohamed[8]

For numbered affiliations see end of article.

**Correspondence to**
Dr Ronald Kiguba;
kiguba@gmail.com

## ABSTRACT

**Introduction** Combination antiretroviral therapy (cART) has massively reduced HIV mortality. However, long-term cART increases the risk of adverse drug reactions (ADRs), which can lead to higher morbidity, mortality and healthcare costs for people living with HIV (PLHIV). Pharmacovigilance—monitoring the effects of medicines—is essential for understanding real-world drug safety. In Uganda, pharmacovigilance systems have only recently been developed, and rates of ADR reporting for cART are very low. Thus, the safety profile of medicines currently used to treat HIV and tuberculosis in our population is poorly understood.

The Med Safety mobile application has been developed through the European Union's Innovative Medicines Initiative WEB-Recognising Adverse Drug Reactions project to promote digital pharmacovigilance. This mobile application has been approved for ADR-reporting by Uganda's National Drug Authority. However, the barriers and facilitators to Med Safety uptake, and its effectiveness in improving pharmacovigilance, are as yet unknown.

**Methods and analysis** A pragmatic cluster-randomised controlled trial will be implemented over 30 months at 191 intervention and 191 comparison cART sites to evaluate Med Safety. Using a randomisation sequence generated by the sealed envelope software, we shall randomly assign the 382 prescreened cART sites to the intervention and comparison arms. Each cART site is a cluster that consists of healthcare professionals and PLHIV receiving dolutegravir-based cART and/or isoniazid preventive therapy. Healthcare professionals enrolled in the intervention arm will be trained in the use of mobile-based, paper-based and web-based reporting, while those in the comparison arm will be trained in paper-based and web-based reporting only.

**Ethics and dissemination** Ethical approval was given by the School of Biomedical Sciences Research and Ethics Committee at Makerere University (SBS-REC-720), and administrative clearance was obtained from Uganda

## STRENGTHS AND LIMITATIONS OF THIS STUDY

⇒ This study will be delivered by researchers, policy-makers, software developers, healthcare providers and consumers to promote medication safety in resource-limited settings.

⇒ We aim to recruit 3820 healthcare professionals from 382 combination antiretroviral therapy sites spread across Uganda; the selected study sites serve 80% of people living with HIV receiving combination antiretroviral therapy.

⇒ A limitation is that the use of the Med Safety mobile app requires a smartphone, but only 16% of Ugandans own a smartphone; limited rollout showed that 7 in 10 healthcare professionals are smartphone owners.

National Council for Science and Technology (HS1366ES). Study results will be shared with healthcare professionals, policymakers, the public and academia.

**Trial registration number** PACTR202009822379650.

## INTRODUCTION

Spontaneous adverse drug reaction (ADR) reporting is the backbone of pharmacovigilance (PV) globally but is plagued by under-reporting.[1–3] Only 3% of Ugandan healthcare professionals (HCPs) reported a suspected ADR to the National Pharmacovigilance Centre (NPC) in 2014.[2] These HCPs cited lack of conventional paper-based ADR forms (paper forms), limited access to web-based forms (web forms) due to the scarcity of internet-wired computers and the lack of feedback to ADR reporters as common barriers to PV. Also, logistical challenges lurk in the

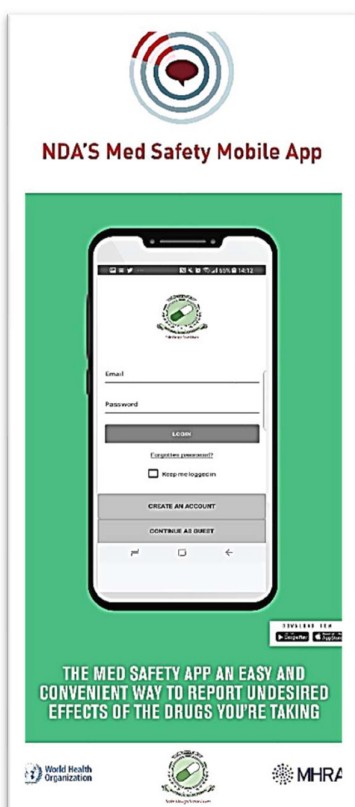

**Figure 1** Med Safety mobile app. MHRA, medicines and healthcare products regulatory agency; NDA, national drug authority.

countrywide distribution and collection of paper forms by Uganda's NPC, which hinders timely data collation and analysis.[2] Ultimately, the implementation of requisite policy actions to safeguard the public from medication-related harm has been delayed.

Innovative interventions are needed to improve ADR reporting, such as the use of mobile applications (mobile apps or apps) to support traditional PV.[4] Mobile apps have had variable success in PV worldwide. A 10-fold increase in the rate of ADR reporting was seen for a medical device with social media engagement in the USA. In contrast, uptake of mobile app reporting has been poor in Europe: studies in the UK, the Netherlands and Croatia showed 1.7, 1.1 and 6.5 app ADR reports per completed month per 1000 app downloads, respectively.[4 5] Uptake was also poor in India: 4.0 app ADR reports per completed month per 1000 app downloads.[6]

Efforts to strengthen PV in Uganda are timely due to the recent rollout of dolutegravir (DTG) as the preferred drug for first-line and second-line combination antiretroviral therapy (cART). DTG is more effective and tolerable and has a higher genetic barrier to developing drug resistance than other antiretrovirals.[7] Rollout began in October 2018 and, by August 2019, over 347 888 people living with HIV (PLHIV) were receiving DTG-based cART.[7] In addition, Uganda is rapidly scaling up isoniazid preventive therapy (IPT) for the prevention of tuberculosis (TB) in PLHIV. Only 16% of PLHIV had been

reached with life-saving IPT by June 2019[8]: an extra 135 711 PLHIV were initiated on IPT by September 2019 after a 100-day scale-up campaign.[7 8] IPT significantly reduces the incidence of active TB, curbing TB-related mortality among PLHIV.[9]

Although DTG is generally well tolerated, it is associated with serious ADRs including hyperglycaemia,[10–13] neuropsychiatric effects (1.7%)[14] and hepatotoxicity (0.1%).[15] DTG is also associated with more common, but less serious, ADRs such as headache, abnormal dreams and abdominal pain, but the impact of these has not been evaluated in real-world settings in developing countries. The rapid rollout of DTG and scale up of IPT will increase the number of PLHIV exposed to these therapies, potentially increasing the incidence of ADRs, including serious ADRs, associated with these medicines. Discontinuation of DTG and/or IPT is recommended if serious ADRs such as jaundice, blurred vision or hyperglycaemia occur.[16] Four in 100 000 PLHIV who receive IPT die due to an IPT-related adverse event.[1]

ADRs can reduce quality of life for PLHIV and increase morbidity, mortality and healthcare costs.[17 18] In 2016, prior to DTG rollout, the NPC attempted to improve spontaneous ADR reporting via a web form. However, they received only 92 online reports versus 290 paper-based ADR reports linked to cART from October 2018 to September 2019. More recently, a PV task force composed of the NPC, the AIDS Control Program and Ministry of Health was set up. The task force established an active drug safety monitoring and management programme to complement the spontaneous ADR reporting system. The NPC received 109 DTG-linked ADR reports from October 2018 to September 2019 from ~348 000 PLHIV. Only 18 IPT-linked ADR reports were received from January 2019 to June 2019 from ~300 000 PLHIV: 2.6 DTG-linked ADR reports per month per 100 000 treated PLHIV and 1.0 IPT-linked ADR report per month per 100 000 treated PLHIV. These rates are barely 5% the known incidence of the ADRs in PLHIV elsewhere.[1 3]

The Med Safety mobile application was developed by the European Union Innovative Medicines Initiative: WEB-Recognising Adverse Drug Reactions. The same technology that produced the Med Safety App was also adopted by European countries including UK's Medicines and Healthcare products Regulatory Agency (MHRA) in July 2015, Netherlands' Lareb in January 2016 and Croatia's HALMED in May 2016. In 2017, the Med Safety App was introduced in Africa (Burkina Faso and Zambia) in partnership with WHO. MHRA adapted Med Safety for low-income and middle-income countries (LMICs) including Uganda with approval from Uganda's National Drug Authority (NDA) (figure 1).[4,19] Med Safety was launched in limited settings in Uganda in February 2020.

Med Safety's platform facilitates easy adoption and low-cost maintenance in LMICs. It is available at no cost for mobile phones and tablets, for Android and iOS, in English. Within the app, ADR reports can be completed

offline and transmitted to NPC when internet connectivity is established. The ADR reporting form has a clear and simple format. App users can browse and view ADR data and may create a 'watch-list' of *medicines* of their own interest to receive personalised news and alerts. Med Safety provides for two-way exchange of medication safety information between NPC and HCPs. This enhanced interaction promotes the involvement of HCPs in PV activities. The anticipated increase in volume of ADR reports in the national PV database could enhance its signal detection potential[20] and contribute to remedial efforts to improve patient safety. The app is also integrated into the WHO Collaborating Centre for International Drug Monitoring – Uppsala Monitoring Centre application programme interfaces (including Vigiflow, VigiAccess and WHODrug).[19] This study seeks to understand whether Med Safety is effective in improving ADR reporting by HCPs if used together with traditional PV methods versus if traditional PV methods are used alone.

## Aims and objectives

Our aim is to assess the feasibility, effectiveness, cost and cost-effectiveness of implementing Med Safety for HCP-driven reporting of ADRs associated with DTG containing cART and IPT for TB prevention in PLHIV in Uganda. Our hypothesis is that the use of Med Safety for ADR reporting by HCPs attending to PLHIV on DTG-based cART and/or IPT will increase the ADR reporting rate[21 22] by at least 25% versus use of existing PV methods alone in 30 months of follow-up at selected cART sites in Uganda. We will test our hypothesis in a pragmatic multicentre open-label cluster-randomised controlled trial whose specific objectives are to:

1. Assess the feasibility and acceptability of Med Safety for ADR reporting by HCPs at selected cART sites in Uganda.
2. Determine the app's effect on the rate of ADR reporting versus traditional PV methods alone.
3. Estimate the app's cost and cost-effectiveness from the provider perspective.

## METHODS AND ANALYSIS
### Participants, interventions and outcomes
#### Study setting

The study will be conducted nationwide at 382 (of 1832 cART sites) high volume accredited cART sites in Uganda. The 382 cART sites serve 80% of the PLHIV on cART in Uganda.[8] This protocol follows the Standard Protocol Items: Recommendations for Interventional Trials 2013 statement (online supplemental file 1).[23]

#### Eligibility criteria

All smartphone-owning HCPs at these sites are eligible including: physicians, medical officers, pharmacists, nurses and midwives, clinical officers, pharmacy technicians and community health workers (lay counsellors and expert clients). Limited rollout showed that 7 in 10 HCPs are smartphone owners. Written informed consent will be sought from eligible HCPs. We expect to enrol 10 HCPs per cART site on average.

#### Intervention arm

HCPs at intervention cART-sites will be introduced to the Med Safety mobile app whether they own a smartphone or not. Mobile app, paper form and web form awareness campaigns including initial face-to-face training, posters/brochures and monthly reminder WhatsApp and mobile phone short messages (SMS) for up to 6 months will be undertaken. Training will be conducted by pharmacists from NPC and Makerere University with expertise in PV. The training teams have harmonised the training schedule to ensure uniform training. Interested HCPs with personal smartphones will be invited and assisted to install the mobile app and trained to use it to report suspected ADRs, with emphasis on DTG-linked and IPT-linked ADRs. HCPs will also be trained and encouraged to use traditional PV methods (paper form and web form).

#### Comparison arm

HCPs at the comparison cART sites will be trained and encouraged to use traditional methods of ADR reporting (paper form and web form). All aspects of the training will be identical to those in the intervention arm except that Med Safety will not be introduced to HCPs in the comparison arm. Reminder WhatsApp and SMS about the paper form and web form will be sent out monthly for up to 6 months.

#### Outcomes

Our primary outcome is number of HCP-reported ADRs per 100 000 person-months of treated PLHIV per study arm. Our secondary outcomes are number of app ADR reports per 1000 app downloads per month of follow-up; causality (by Naranjo Scale and Liverpool Causality Assessment Tool)[24 25]; seriousness as per the WHO definitions (threatens life, ie, leads to or prolongs hospitalisation, causes incapacitation or death); ADR outcome; cost per ADR report; cost per additional ADR report; and cost per additional avoidable serious ADR report.

#### Participant timeline

There are 15 regional referral hospitals each with a catchment of ~25 cART sites. Each team of three research assistants requires 4 weeks to enrol HCPs in one regional catchment. Follow-up at each cART site begins after its enrolment and lasts 30 months. The 30-month follow-up period is considered adequate to monitor durability of the real-life impact of the app on ADR reporting. Four additional months will be required to wrap-up the study, thus, 36 months overall (table 1).

#### Sample size

To estimate the number of cART sites required per arm for an effect size of 25%,[26] we assume power of 80% at the 95% confidence level, mean of 1.0 IPT-linked ADR/month/100 000 treated PLHIV (0.00001) and SD of 1.179 IPT-linked ADRs/month/100 000 treated PLHIV

**Table 1** Schedule of enrolment, interventions and assessments

| | Study period | | | | | | |
|---|---|---|---|---|---|---|---|
| | Baseline | Follow-up schedule | | | | | Close-out |
| Procedures | $(t_0)$ | $t_0$+3 months | $t_0$+6 months | $t_0$+12 months | $t_0$+15 months | $t_0$+30 months | $t_0$+36 months |
| **Enrolment** | | | | | | | |
| Eligibility screen | × | | | | | | |
| Informed consent | × | | | | | | |
| Allocation to study arm | × | | | | | | |
| **Interventions** | | | | | | | |
| Mobile app arm | x | x | x | x | x | x | |
| Comparison arm | x | x | x | x | x | x | |
| **Assessments** | | | | | | | |
| Quantitative survey | x | | | x | | | |
| Mobile app data | x | x | x | x | x | x | |
| Economic cost data | x | x | x | x | | | |
| Interim data analysis | | | | | x | | |
| Qualitative study | x | | | x | | | |
| Endline data analysis | | | | | | | x |

(0.00001179) (SD was computed based on the monthly IPT-linked ADR reports submitted to NPC for 1 year from October 2018 to September 2019), cluster size of 10 HCPs and coefficient of variation of 0.25,[26] which yields ~114 cART sites per trial arm or 228 in both study arms. A minimum of 189 cART sites per arm or 378 in both study arms (66% increment) will be required to cater for the following limitations: (1) the large proportion of HCPs who do not own smartphones of ~50%–60% as previously estimated[21 22] – and is 26% (95% CI of 23% to 30%) from limited rollout of RCT; (2) refusal to participate by eligible HCPs of ~5% as previously estimated[27] – and is 0% (95% CI of 0% to 2%) from limited rollout of RCT; (3) loss to follow-up due to the relatively long follow-up period of 30 months (~10%); and (4) contamination due to information sharing between study arms through social media (~30%).[28] We shall enrol 382 cART sites (191 per study arm) to cater for other unforeseen limitations; thus, recruiting up to 3820 HCPs (1910 HCPs per trial arm). Limited rollout of the RCT suggests a cluster size of 7.2

(629/87) HCPs. We shall retain the average cluster size of 10 HCPs because the RCT has not yet enrolled from the central region of Uganda (with the country's capital city) where cART sites tend to be much larger with more HCPs per site than the more rural eastern, western and northern regions.

## Assignment of interventions
### Unit of randomisation
This unit is a cluster, defined as an cART site located at each of the 382 prescreened health facilities and consisting of HCPs and DTG/IPT-treated PLHIV. The use of cART sites as clusters minimises cointervention, duplication of reporting and organisational challenges and provides valid PLHIV population denominators for analysis of the outcomes.

### Concealment of sequence generation
Using the sealed envelope software, an independent Biostatistician from the Clinical Epidemiology Unit at Makerere University, who will not participate in the subsequent data analyses, generated the simple randomisation sequence.

### Randomisation
The prescreened cART sites will be assigned by research assistants to the intervention and comparison arms, figure 2, using an electronically generated simple randomisation sequence.

### Blinding
PLHIV, HCPs, PV assessors of the ADR reports submitted to NPC and the biostatistician who will analyse the data will be blinded to the allocation of cART sites. The research assistants will not be blinded to the allocation of cART sites due to the nature of the intervention.

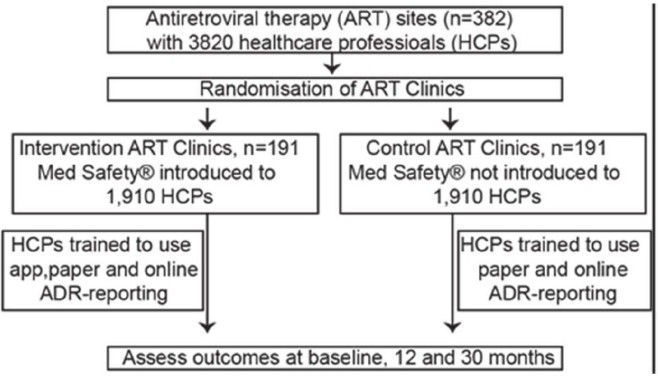

**Figure 2** Flow diagram for cluster-randomised trial at full rollout. ADR, adverse drug reaction.

## Data collection
### Baseline survey
Consented HCPs will complete an interviewer-administered electronic questionnaire using the Open Data Kit software to record participants' characteristics and details on ADR reporting. Details of the questionnaires are provided in (online supplemental file 2).

### Mobile app data
The app is hosted by NDA where the NPC is located, which is also the preferred source of ADR risk information for users of the mobile app. HCPs with smartphones will submit reports that capture details on patient sociodemographics, suspected medicine, other concurrent medicines, suspected ADR and medical history. The NPC transmits alerts via the mobile app on emerging medication safety issues to users of the app.

### Paper form and web form
The paper form and web form details will be distributed at all enrolled cART-sites. PV focal persons based at the cART-sites and NDA regulatory officers will routinely collect the paper forms and forward them to NPC for data capture in the national PV database and central analysis, prior to onward submission to the WHO database, VigiBase. HCPs also have the option to submit online ADR reports using the web form.

### Economic cost data
Cost data will be collected by an appropriately trained RA. We will collect *setup costs* such as app development costs, equipment costs, cost of vehicles and training costs, and *running costs* such as airtime, internet data, app maintenance, personnel costs, buildings and space if any, transport (fuel), stationary, brochures, costs of registers and airtime/reminders. Both *setup* and *1-year running costs* will be collected from the provider perspective.

### Feasibility and acceptability of the app
Prior to trial implementation, we will conduct a baseline qualitative study to gauge the acceptability and feasibility of introducing the app to HCPs. We will conduct three to five focus group discussions each with five to eight HCPs and 20–30 in-depth interviews in a random 6% of cART-sites (~12) in the intervention arm. During trial implementation, we will document the refusal and failure rates to instal the app among consented HCPs in the intervention arm. App users will be asked to report their experiences to gauge app feasibility, assess user satisfaction and identify potential revisions to the app. We will gauge acceptability of the app based on whether users can recommend the app to other HCPs to report suspected ADRs to NPC.

## Data management
### Training of research assistants
Both the initial and ongoing trainings focus on the key concepts of PV, good clinical practice, human subject protection, randomisation, risk profiles of cART and IPT

and techniques for collecting high-quality survey data among others.

### Data quality assurance
The PV database quality manager at NDA will be trained and empowered by the research team to embed data quality management in the national PV database.

### Data protection
All ADR data are processed and securely stored by NPC. Paper forms are manually entered into the PV database by authorised NPC staff, while online data are electronically transmitted into VigiFlow and assessed before transfer into VigiBase. Researchers can request anonymised safety data for research purposes only. Paper-based questionnaires are stored under lock and key and accessed only by authorised project staff. Electronic data are password protected and used for research purposes only. Data will be delinked from HCP identifiers prior to analysis.

### Compliance with allocation
Prior to data anonymisation, an NPC officer not involved in the routine assessment of submitted ADR reports will code these reports according to the allocated trial arm of the site where the report originates. Our team will use the codes to assess the extent of cross-over.

## Data analysis
### Success of randomisation
Baseline characteristics (sociodemographics of HCPs, pattern of smartphone ownership by HCPs at cART sites, geographical location of cART-site, nature and level of health facility, number and cadres of HCPs, number of PLHIV and use of DTG and IPT) will be compared in both study arms to establish if randomisation was successful.

### Quantitative data
Analysis of results will be reported according to Consolidated Standards of Reporting Trials guidelines and performed on both intention-to-treat and per protocol bases. All ADR data received from the ~382 cART sites during the study period will be downloaded from VigiBase and anonymised by the PV manager at NPC and subsequently provided to the research team. Duplicate ADR reports will be identified and excluded from formal statistical analyses. The randomisation code will be broken after data analysis.

### Unit of analysis
This unit will be an ADR site in keeping with the cluster-randomised design.

### Analysis of missing or lost clusters
We will compare baseline characteristics of individuals in the lost clusters with the characteristics of individuals in the clusters that will have completed follow-up. If we find no significant differences between the clusters, we will conclude that our results do not include any differential

misclassification. If there are differences, however, we will report this finding and discuss its implications.

### Descriptive statistics

We shall determine the frequencies of deduplicated HCP-reported ADRs per 100 000 treated PLHIV per cART site together with the time from ADR onset to VigiBase registration.[29]

### Effectiveness

Descriptive data will be aggregated to obtain the number of HCP-reported ADRs per 100 000 treated PLHIV per trial arm and compared at cluster level using the Student's t-test or Wilcoxon rank-sum test, as appropriate, to determine if the app significantly increases ADR reporting. We will assess if the introduction of Med Safety increases the rate of ADR reporting by HCPs to NPC versus the use of existing PV methods (paper and online) alone during 30 months of follow-up and effect size of 25% at 5% level of statistical significance and power of 80%.

### Multivariable analysis

Models will use the number of ADR reports as the outcome. Hierarchical models (level 1: cART sites; level 2: HCPs nested within cART sites; level 3: individuals nested within HCPs) will be fitted using maximum log-likelihood, considering intracluster correlations, using mixed models and generalised estimating equations models to control for potential confounders and effect modifiers identified at baseline assessment if fair randomisation fails. Subgroup analyses will also be performed.

### Feasibility and acceptability of the app

We will compute the refusal rates and detail the reasons for failed mobile app installation by consented HCPs in the intervention arm. Qualitative data on the acceptability and barriers/facilitators of using the mobile app by HCPs will be analysed using thematic analysis by employing NVivo V.10 software.

### Cost and cost-effectiveness

We will estimate unit *setup* and *running costs* for both the introduction of Med Safety in addition to existing PV methods and the use of existing PV methods alone. Costs per ADR report submitted to NPC will be computed (overall; per study arm; per ADR attribute, eg, seriousness, avoidability, causality, etc). We will estimate cost-effectiveness of introducing Med Safety by calculating the incremental cost-effectiveness ratio, which is the cost per additional ADR report submitted to NPC.

### Monitoring
#### Monitoring and evaluation mechanism

The research team will monitor the project's performance based on the research activities and report progress to the Project Steering Committee, which will sit (online) every 6 months. During monitoring, the research team will continually collect and analyse information on the project's ongoing research activities. The steering committee will provide independent external evaluation of the project based on the 6 monthly reports from the research team.

### Stopping rules

A Data Safety Monitoring Board is in place. Midterm interim analysis will be conducted at p<0.01 to assess if the app is substantially better than a priori estimates. The study will otherwise stop when all 382 cART sites have been enrolled and followed up for 30 months. A protocol for patient care is in place at each cART site to evaluate and manage PLHIV who develop suspected ADRs linked to cART and IPT.

### Ethics and dissemination

We obtained ethical approval for the study from the School of Biomedical Sciences Research and Ethics Committee at Makerere University College of Health Sciences (SBS-REC-720), and subsequently undertook research registration with the Uganda National Council for Science and Technology (HS1366ES). Administrative clearance is being obtained from participating cART sites and written informed consent sought from participating HCPs (online supplemental file 3).

### Dissemination strategies

During the study, limited communication tailored specifically for each study arm will be made by the study team to limit the effect of contamination.

### Engaging healthcare providers

We shall employ video conferencing, webinars and face-to-face engagements to deliver continuing professional education sessions to HCPs on ADR risk assessments for PLHIV prior to the initiation of DTG regimens and IPT and the detection, management and reporting of ADRs for PLHIV already receiving DTG/IPT. Video conferencing is preferred to minimise the spread of COVID-19. If virtual meetings are not feasible, then face-to-face meetings will be held in full observance of the Ministry of Health's guidelines for limiting the spread of COVID-19: social distancing, use of face masks and handwashing, especially for regional trainings of HCPs in remote areas where the internet might not be accessible and during the enrolment of study sites. The NPC in concert with Ministry of Health developed PV training materials for active drug safety monitoring and management of ADRs to DTG/IPT. We shall adapt and update these materials with the results of our study for academic training and also tailor them for healthcare service delivery trainings to be conducted nationwide in partnership with NPC and Ministry of Health.

### Reaching healthcare providers, policymakers and drug regulators

This work underpins the wider rollout of Med Safety to improve PV across Uganda and beyond. This study is being implemented in direct partnership with cART sites, Uganda's NDA, which is the PV coordinating body in Uganda, UK's MHRA and WHO. Therefore, the study

results will be readily available for immediate use to inform policy and practice by these key stakeholders. The NDA and MHRA (the software developers) will identify the barriers to the app's implementation and mitigate them as well as the facilitators and exploit them. Uganda's experience could be exploited to scale up the app in other low-income and middle-income countries.

### Reaching researchers

In addition to social media and webinars, as mentioned previously, we aim to reach researchers through traditional methods. We shall present our work at conferences including: Joint Annual Scientific Health Conference, Makerere University College of Health Sciences; International Society of Pharmacovigilance Annual Meeting; Conference on Retroviruses and Opportunistic Infections; International AIDS Society Conference, Royal Society of Tropical Medicine and Hygiene Annual Meeting, among others. We expect to produce at least six open access publications from this work by the end of this study. We shall target journals including: Lancet Global Health; Drug Safety; Pharmacoepidemiology and Drug Safety, Journal of the International AIDS Society; Journal of Antimicrobial Chemotherapy; and Journal of Acquired Immune Deficiency Syndromes, etc.

### Project impact

Improving the reporting of suspected ADRs via digital PV is the first step towards boosting the volume of safety data available for robust signal detection analyses, improving understanding of the risk profiles of medicines and preventing future occurrence of avoidable ADRs. Improved PV translates into better recognition of serious ADRs and, ultimately, safer patient care. This work underpins future proposals to understand the risk factors for ADRs in PLHIV.

### Protocol amendments

Protocol modifications that may impact on implementation of the study and affect patient safety including changes of study objectives, study design, study population, sample sizes, study procedures or significant administrative aspects will require a formal protocol amendment. Such amendment will be agreed on by the study investigators and approved by the School of Biomedical Sciences Research & Ethics Committee prior to implementation.

### Patient and public involvement

Patients were not involved in designing this study and will not be directly involved in the conduct, reporting and dissemination of the research. However, HCPs were involved in refining the study tools and will participate in conducting, reporting and disseminating the research.

**Author affiliations**
[1]Department of Pharmacology and Therapeutics, College of Health Sciences, Makerere University, Kampala, Uganda
[2]Department of Epidemiology & Biostatistics, College of Health Sciences, Makerere University, Kampala, Uganda
[3]National Pharmacovigilance Centre, National Drug Authority, Kampala, Uganda
[4]AIDS Control Programme, Ministry of Health, Kampala, Uganda
[5]MRC/UVRI and LSHTM Uganda Research Unit, Entebbe, Uganda
[6]Vigilance and Risk Management of Medicines, Medicines and Healthcare Products Regulatory Agency, London, UK
[7]Clinical Epidemiology Unit, Makerere University College of Health Sciences, Kampala, Uganda
[8]MRC Centre for Drug Safety Science and Wolfson Centre for Personalised Medicine, Institute of Systems, Molecular and Integrative Biology (ISMIB), University of Liverpool, Liverpool, UK

**Acknowledgements** UK's Medicines and Healthcare products Regulatory Agency and Uganda's National Drug Authority provided technical and logistical support during the planning and implementation of this project.

**Contributors** RK, HBN, VB, CK and MP designed the study. RK drafted the manuscript. RK, NM, RS, HBN, VN, CK, KRK, PT, KH, CK and MP critically reviewed and revised the final version of the manuscript. All authors read and approved the final manuscript.

**Funding** This project is supported by the Medical Research Council (MR/V03510X/1) and Makerere University Research & Innovations Fund (N/A).

**Disclaimer** The funders had no role in study design, data collection and analysis, decision to publish or preparation of manuscripts.

**Competing interests** None declared.

**Patient and public involvement** Patients and/or the public were not involved in the design, or conduct, or reporting, or dissemination plans of this research.

**Patient consent for publication** Not applicable.

**Provenance and peer review** Not commissioned; externally peer reviewed.

**ORCID iDs**
Ronald Kiguba http://orcid.org/0000-0002-2636-4115
Ronald Ssenyonga http://orcid.org/0000-0002-2409-3593

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
