## [Reviewer comments · BMJ Open]

ARTICLE DETAILS

TITLE (PROVISIONAL)	Effectiveness of the Med Safety mobile application in improving adverse drug reaction reporting by healthcare professionals in Uganda: a protocol for a pragmatic cluster-randomized controlled trial
AUTHORS	Kiguba, Ronald; Mwebaza, Norah; Ssenyonga, Ronald; Ndagije, Helen; Nambasa, Victoria; Katureebe, Cordelia; Katumba, Kenneth; Tregunno, Phil; Harrison, Kendal; Karamagi, Charles; Scott, Kathryn A.; Pirmohamed, Munir

VERSION 1 – REVIEW

REVIEWER	Lopes, Luciane Universidade de Sorocaba, Pharmaceutical Science
REVIEW RETURNED	17-Apr-2022

GENERAL COMMENTS	This is well design and written protocol. I have only a few suggestions: a) please conceptualize better feasibility and acceptability in the methods section; b) please, include information in the Methods section on how will be the analysis of missing/lost clusters and individuals assessed; c) It needs a better explanation how the blind of all persons will be keeping during the follow-up.
--

REVIEWER	Janković, Slobodan University of Kragujevac
REVIEW RETURNED	08-May-2022

GENERAL COMMENTS	The manuscript is describing a protocol of interventional clinical trial aiming to investigate effectiveness of a mobile application designed to promote ADR reporting. The topic of the trial is novel, and prospective results would have great practical significance, since underreporting of ADR is affecting almost all countries and regions. Language of the manuscript and the style of writing are good. Randomization technique, definition of study population, sample size, statistical plan, study variables are all appropriate and reliably planned. There are no ethical concerns. However, there is one major study design issue that should be resolved: - experimental and control group should be treated in the same way, except by the intervention itself, in order to obtain an insight in the efficacy of the intervention free of confounders. However, the authors did not ensure that training of the physicians in regard to the ADRs and their motivation to report ADRs should be exactly the same in both groups. The outcomes of the study are extremely sensitive to training and motivation of the physicians. The authors should elaborate in much more detail how they will provide for the same training and the same motivation techniques and intensity in both experimental and control group.
---

REVIEWER	Mestres, Conxita School of Health Sciences Blanquerna
REVIEW RETURNED	15-May-2022

GENERAL COMMENTS	Spontaneous adverse drug reactions reporting is a very powerful tool in order to have knowledge of the frequency and seriousness of ADR after registration. However, the great problem of underreporting diminishes the effectiveness of this method of Pharmacovigilance. The use of apps in this process can be very useful. Therefore, the idea of the authors of implementing this tool in a country as Uganda are good news. The protocol is very exhaustive and precise. There are some little things I would like to point out:  - Data collection, page 7 line 49. The authors explain that paper-forms would be collected and forwarded to NPC. Who would collect them? How the authors can assure that the forms would reach its destination? - In the protocol the authors refer to Health Care Practitioners that could report ADR. They did not specify in the protocol which HCP can or are allowed to report ADR. In the supplementary file that contains the questionnaire for HCP, there is a list of them, but I think that it is important to state specifically in the protocol the details of what HCP can report.
---

VERSION 1 – AUTHOR RESPONSE

Reviewer 1

Comment 1: This is a well designed and written protocol.

Response 1: Thank you.

Comment 2: Please conceptualize better the feasibility and acceptability in the methods section.

Response 2: We have deleted the sub-section titled 'Follow-up survey'. The sub-section titled 'Feasibility and acceptability of the app' is revised as follows: "Prior to trial implementation, we will conduct a baseline qualitative study to gauge the acceptability and feasibility of introducing the app to HCPs. We will conduct 3-5 Focus Group Discussions each with 6-8 HCPs and 20-30 in-depth interviews in a random 6% of cART-sites (~12) in the intervention arm. During trial implementation, we will document the refusal and failure rates to install the app among consented HCPs in the intervention arm. App-users will be asked to report their experiences to gauge app-feasibility, assess user-satisfaction, and identify potential revisions to the app. We will gauge acceptability of the app based on whether users can recommend the app to other HCPs to report suspected ADRs to NPC." (Page 8, lines 14-21).

Comment 3: Please, include information in the Methods section how the analysis of missing/lost clusters and individuals will be assessed.

Response 3: We have added: "We will compare baseline characteristics of individuals in the lost clusters with the characteristics of individuals in the clusters that will have completed follow-up. If we find no significant differences between the clusters, we will conclude that our results don't include any differential misclassification. If there are differences, however, we will report this finding and discuss its implications." (Page 8, lines 48-52).

Comment 4: ...how will the blinded persons be kept blinded during follow-up?

Response 4: The recruitment of cART-sites as clusters minimizes contamination (Page 7, lines 23-

24). During study implementation, limited communication tailored specifically for each study arm will be made by the study team to limit the effect of contamination (Page 10 lines 8-9). However, this is a pragmatic trial conducted in routine practice conditions so we will not be able to completely prevent contamination during follow-up. The study has anticipated up to 30% level of contamination, which has been factored into sample size estimation (page 7, lines 13-14). We will also assess the actual extent of contamination at data analysis (Page 8, lines 45).

Reviewer 2

Comment 5: The topic of the trial is novel, and prospective results would have great practical significance, since underreporting of ADR is affecting almost all countries and regions.

Response 5: Agreed. Thank you.

Comment 6: Language of the manuscript and the style of writing are good. Randomization technique, definition of study population, sample size, statistical plan, study variables are all appropriate and reliably planned. There are no ethical concerns.

Response 6: Thank you.

Comment 7: However, there is one major study design issue that should be resolved:

- experimental and control group should be treated in the same way, except by the intervention itself, in order to obtain an insight in the efficacy of the intervention free of confounders. However, the authors did not ensure that training of the physicians in regard to the ADRs and their motivation to report ADRs should be exactly the same in both groups. The outcomes of the study are extremely sensitive to training and motivation of the physicians. The authors should elaborate in much more detail how they will provide for the same training and the same motivation techniques and intensity in both experimental and control group.

Response 7: An additional sentence is added to the sub-section titled 'Comparison arm': "All aspects of the training will be identical to those in the intervention arm except that Med Safety will not be introduced in the comparison arm." (Page 5, line 52 & Page 6 line 1-2).

Reviewer 3

Comment 8: Spontaneous adverse drug reactions reporting is a very powerful tool in order to have knowledge of the frequency and seriousness of ADR after registration. However, the great problem of underreporting diminishes the effectiveness of this method of Pharmacovigilance.

The use of apps in this process can be very useful. Therefore, the idea of the authors of implementing this tool in a country as Uganda are good news.

Response 8: Agreed. Thank you.

Comment 9: The protocol is very exhaustive and precise.

Response 9: Thank you.

Comment 10: Data collection, page 7 line 49. The authors explain that paper-forms would be collected and forwarded to NPC. Who would collect them? How will the authors ensure that the forms reach their destination?

Response 10: The sentence has been revised as follows: "Pharmacovigilance focal persons based at the cART-sites and NDA regulatory officers will routinely collect the paper-forms and forward them to NPC for data capture in the national PV database and central analysis, prior to onward submission to the WHO database, VigiBase." (Page 8, lines 1-3).

Comment 11: In the protocol the authors refer to Health Care Practitioners that could report ADR.

They did not specify in the protocol which HCP can or are allowed to report ADR. In the supplementary file that contains the questionnaire for HCP, there is a list of them, but I think that it is important to state specifically in the protocol the details of what HCP can report.

Response 11: The manuscript is updated to include the HCP cadres: “All smartphone-owning HCPs at these sites are eligible and include physicians, medical officers, pharmacists, nurses and midwives, clinical officers, pharmacy technicians and community health workers (lay counsellors and expert clients)” (Page 5, lines 35-37).

Comment 12: *Please revise the ‘Strengths and limitations of this study’ section of your manuscript (after the abstract). This section should contain up to five short bullet points, no longer than one sentence each, that relate specifically to the methods. The novelty, aims, results or expected impact of the study should not be summarised here.

Response 12: Revised accordingly. See Page 3.